# Persistence of Anti-SARS-CoV-2 Antibodies Six Months after Infection in an Outbreak with Five Hundred COVID-19 Cases in Borriana (Spain): A Prospective Cohort Study

Salvador Domènech-Montoliu [1], Joan Puig-Barberà [2], Maria Rosario Pac-Sa [3], Paula Vidal-Utrillas [4], Marta Latorre-Poveda [1], Alba Del Rio-González [4], Sara Ferrando-Rubert [4], Gema Ferrer-Abad [4], Manuel Sánchez-Urbano [1], Laura Aparisi-Esteve [5], Gema Badenes-Marques [1], Belén Cervera-Ferrer [1], Ursula Clerig-Arnau [1], Claudia Dols-Bernad [6], Maria Fontal-Carcel [7], Lorna Gomez-Lanas [1], David Jovani-Sales [5], Maria Carmen León-Domingo [8], Maria Dolores Llopico-Vilanova [1], Mercedes Moros-Blasco [5], Cristina Notari-Rodríguez [1], Raquel Ruíz-Puig [1], Sonia Valls-López [1] and Alberto Arnedo-Pena [3,9,10,*]

1   Emergency Service Hospital de la Plana, 12540 Vila-real, Spain; domenech_salmon@gva.es (S.D.-M.); martalapo@hotmail.com (M.L.-P.); manu.msu@gmail.com (M.S.-U.); gemabamar@hotmail.com (G.B.-M.); belencerveraferrer@hotmail.com (B.C.-F.); ursuclerig@gmail.com (U.C.-A.); lornagl78@gmail.com (L.G.-L.); llopivila@hotmail.com (M.D.L.-V.); notari_cri@gva.es (C.N.-R.); raquelruizpuig@gmail.com (R.R.-P.); Sonia.valls.lopez@gmail.com (S.V.-L.)
2   Vaccines Research Area FISABIO, 46020 Valencia, Spain; jpuigb55@gmail.com
3   Public Health Center, 12540 Castelló de la Plana, Spain; charopac@gmail.com
4   Health Centers I and II, 12540 Borriana, Spain; vidalutrillaspaula@gmail.com (P.V.-U.); delrio_alb@gva.es (A.D.R.-G.); sfr1812@gmail.com (S.F.-R.); gferrer@uji.es (G.F.-A.)
5   Carinyena Health Center, 12540 Vila-real, Spain; lauraaparisiesteve@gmail.com (L.A.-E.); jovasal1987@gmail.com (D.J.-S.); mercedesmb1094@hotmail.com (M.M.-B.)
6   Health Center, 12540 Onda, Spain; claudiadb1294@hotmail.com
7   Health Center, 12540 La Vall d'Uixó, Spain; fontalcarcel93@hotmail.com
8   Villa Fátima School, 12540 Borriana, Spain; carmendole@hotmail.com
9   Department of Health Science, Public University Navarra, 31006 Pamplona, Spain
10  Epidemiology and Public Health (CIBERESP), 28029 Madrid, Spain
*   Correspondence: arnedo_alb@gva.es or albertoarnedopena@gmail.com; Tel.: +35-622-573979

**Abstract:** In March 2020, several mass gathering events were related to the *Falles* festival in Borriana (Spain), resulting in a 536 laboratory-confirmed COVID-19 cases outbreak among participants. This article estimates anti-SARS-CoV-2 antibodies persistence six months after and factors associated with antibody response. A prospective population-based cohort study was carried out by the Public Health Centre of Castellon and the Emergency and Clinical Analysis and Microbiology Services of Hospital de la Plana in Vila-real. In October 2020, a seroepidemiologic study was used to estimate the persistence of anti-SARS-CoV-2 antibodies against nucleocapsid protein (N) by an electrochemiluminescence immunoassay (ECLIA) was implemented. We enrolled 484 (90.2%) of the 536 members of the initial outbreak cohort and detected persistent antibodies in 479 (99%) without reinfection episodes. Five participants had a negative antibody test. Factors associated with a negative result were a lower body mass index (BMI), and less contact with other COVID-19 cases. Among the 469 participants with two ECLIA tests, 96 (20.5%) had an increase of antibodies and 373 (79.5%) a decline. Increased antibodies were associated with older age, higher BMI, more severe illness, and low current smokers. Our results show that after a COVID-19 infection, a high proportion of cases maintain detectable anti-SARS-CoV-2 antibodies.

**Keywords:** SARS-CoV-2; COVID-19; antibodies; cohort; population-based; body mass index; ECLIA

## 1. Introduction

At the time of writing, severe acute respiratory syndrome coronavirus 2 (SARS-CoV-2) has caused 160,457,476 confirmed cases and 3,331,604 deaths globally [1]. One of the

main unknowns is the duration of immunity elicited after coronavirus disease (COVID-19) in recovered patients. Other aspects of interest are the protection against SARS-CoV-2 reinfection, and factors related to developing persistent SARS-Cov-2 immunity. Serologic surveys are needed to gather evidence on the persistence and protection of the humoral immunity to SARS-CoV-2 infection [2,3], as well as the study of the cellular immunity mediated by T cells [4,5].

In the first days of March 2020, several mass gathering events (MGEs) for the *Falles* festival took place in Borriana, a city of 35,000 inhabitants in Castellon (Spain), and a COVID-19 outbreak ensued. During March–June 2020, an epidemiologic study whit a serologic survey of this outbreak found 570 COVID-19 cases (536 laboratory-confirmed and 34 with clinical and epidemiologic criteria), 13 admissions and one death, among 1338 participants in the MGEs; the attack rate (AR) was 42.6%.

The follow-up of a representative sample of the COVID-19 cases through a serologic survey to determine SARS-CoV-2 antibodies would offer the opportunity to know the duration of immunity after COVID-19. Accordingly, our objective was to conduct a seroepidemiological study, six months after the first study, to estimate the persistence of anti-SARS-CoV-2 antibodies among MGEs participants who suffered a laboratory-confirmed COVID-19 infection and the potential factors associated with the persistence and intensity of the immune response.

## 2. Materials and Methods

The study, a prospective population-base cohort study, was carried out by the Public Health Centre of Castellon and the Emergency and Clinical Analysis and Microbiology Services of Hospital de la Plana in Vila-real. We invited all the subjects who tested positive for SARS-CoV-2 in the outbreak of the MGEs of *Falles* festival to participate in a second seroepidemiological study in October 2020.

The tests used to ascertain SARS-CoV-2 infection in the first study were: Qualitative Electrochemiluminescence immunoassay (ECLIA) (Elecsys® Anti-SARS-CoV-2, Roche Diagnostics) [6], in 514 subjects. Lateral flow immunochromatographic assay (LFIC) (Healgen Scientific LLC for COVID-19 IgG/IgM rapid test cassette [7], in 15 subjects. Reverse-transcriptase polymerase chain reaction (RT-PCR), including LightMix® Modular Sarbecovirus E-gene with the LightCycler® 480 II system [8], in 39 subjects. In 32 subjects, we obtained both ECLIA and PCR results.

The outbreak cohort members with a laboratory-confirmed COVID-19 test were 536 cases from the first serologic survey between March and June 2020. The second serologic survey to determine anti-SARS-CoV-2 antibodies by ECLIA was implemented during October 2020, and 484 of the 536 members of the cohort took part (90.3%). Overall, 469 members had two ECLIA determinations after the second serologic survey (Figure 1).

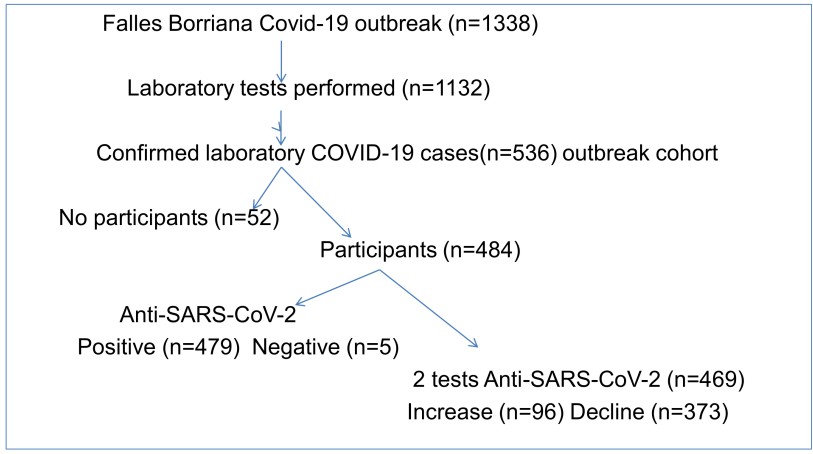

**Figure 1.** Flow chart of Borriana COVID-19 outbreak cohort.

The ECLIA test, already described above, uses a modified double-antigen sandwich immunoassay using recombinant SARS-CoV-2 nucleocapsid protein (N) [6]. The test results are reported as numeric values, considering a reactive cutoff index (COI) > 1.0 as a positive result and a COI < 0.1 as a negative result.

We explore factors associated with the immune response, comparing the geometric mean of total anti-SARS-CoV-2 antibodies of participants in the first and second serologic surveys and the change in magnitude between tests as an increase or a decline. The first blood samples were not available for a simultaneous analysis with the second samples. We also explored the potential associations [9–12] with COVID-19 of ABO blood groups, ascertained by the gel test (ID-Card ABO/RhD, DiaMed GmbH, Bio-Rad Laboratories Switzerland) [13] and 25-hydroxy vitamin D [25(OH)D] levels, determined by electrochemiluminisence-based assay (Elecsys vitamin D total II Roche Diagnostic, Germany) [14].

Health staff of the Hospital de la Plana, and the health centres of Borriana, Vila-real, Onda, and La Vall d'Uixo conducted a telephone questionnaire survey in October 2020, asking the participants about their health situation, the occurrence of disease in the last six months, medical assistance, evolution of COVID-19 disease, sequelae, SARS-CoV-2 reinfections, and subsequent exposures. We obtained information about other potential factors associated with the immune response, such as age, sex, weight, height, body mass index (BMI) ($kg/m^2$), occupation, level of physical exercise, smoking habits, consumption of alcohol, chronic illness and COVID-19 exposures, either in the household or in other settings from the questionnaire survey during May-June 2020.

*Statistical Analysis*

We used the Chi2 test and Fisher exact test for qualitative variables, Kruskal-Wallis test for quantitative variables, and Wilcoxon matched-pairs signed-rank test and equality of median for matched pairs test to compare the changes in anti-SARS-CoV-2 antibodies between the two serologic surveys. Positive or negative detection of anti-SARS-CoV-2 antibodies in the second respect to the first survey was the dependent variable. We estimated the associations between this variable and several factors by relative risks (RR) through univariate and multivariate (adjusted RR) Poisson regression, robust or exact according to models' conditions, with 95% confidence intervals (CI). Besides, we analysed the increase and decline in anti-SARS-CoV-2 antibody arbitrary units following the same statistical approach. As Vitamin D was only measured in the second serologic survey, we did not include it in multivariate models. After a review of SARS-CoV-2 medical literature, we studied the potential confounding factors by Directed Acyclic Graphs (DAGs) [15] using the DAGitti 3.0 program [16]. With a sample size of 454 cohort members, the study had a power of 80% and alpha error of 5% to detect at least a 2% change in the fraction of positives between the first and second serologic surveys. We used Stata ® version 14 statistical program for all calculations and estimates.

The following of the study's cohort was a part of the public health surveillance as a prolongation of the COVID-19 outbreak control measures, and the response of the COVID-19 pandemic. It was exempt from the Ethics Review Board approval's protocol according to the Spanish legislation. The study was approved by the director of the Public Health Center of Castellon and the management of the Health Department of la Plana. All participants or the parents of minors provided the informed written consent to be included in the study.

## 3. Results

A total of 484 (90.2%) of 536 subjects accepted to participate (Figure 1) (Table 1). We did not observe significant differences between participants and no-participants, except for MGEs assistance, which was lower in non-participants ($p < 0.001$).

From 484 participants in the outbreak cohort, persistent antibodies were detected in 479 (99%), and reinfection episodes were detected. Only five of the 484 participants (1.0%) had a negative anti-SARS-CoV-2 antibody in the second determination.

**Table 1.** Participants and non-participants characteristics. COVID-19 Borriana cohort second SARS-CoV-2 serologic survey, October 2020.

| Variables | Participant N = 484 (%) | No-Participants N = 52 (%) | *p*-Value |
|---|---|---|---|
| Female | 301 (62.2) | 32 (61.5) | 1.000 |
| Age mean ± Standard Deviation | 37.2 ± 17.1 | 33.5 ± 16.7 | 0.064 |
| 0–24 years | 143 (29.5) | 19 (36.5) | |
| 25–44 | 157 (32.4) | 17 (32.7) | |
| 45–64 | 166 (34.3) | 16 (30.8) | 0.483 |
| 65 and over | 18 (3.7) | 0 (0.0) | |
| Body mass index [1] Kg/m$^2$ | | | |
| <18.0 | 41 (8.5) | 6 (11.5) | |
| 18.0–24.9 | 210 (43.7) | 24 (46.2) | |
| 25.0–29.9 | 148 (30.8) | 13 (25.0) | 0.747 |
| ≥30.0 | 85 (17.7) | 9 (17.3) | |
| Occupation I-II [2,3] | 145 (30.1) | 16 (30.7) | 1.000 |
| Current smoker [4] | 65 (13.9) | 12 (23.1) | 0.221 |
| Physical exercise | 389 (80.4) | 24 (46.2) | 0.075 |
| Alcohol beverages [5] | 108 (23.0) | 10 (19.2) | 0.604 |
| Chronic illness [6] | 166 (34.6) | 14 (26.9) | 0.285 |
| COVID-19 | | | |
| Family with COVD-19 case [7] | 303 (62.7) | 31 (59.6) | 0.554 |
| Probable contact COVID-19 case [8] | 390 (81.8) | 39 (75.0) | 0.349 |
| Assistance Mass Gathering events ≥ 2 and over [9] | 295 (61.0) | 16 (30.8) | 0.000 |
| Hospitalisations | 9 (1.9) | 3 (5.8) | 0.101 |
| PCR positive | 26 (5.4) | 13 (25.0) | NC [10] |
| Asymptomatic | 54 (11.2) | 10 (19.2) | 0.111 |
| Medical consultation | 208 (43.0) | 25 (48.1) | 0.556 |

[1] Missing information for 3 participants; [2] Occupation groups I-II: Professional, managerial and technical occupations. Groups III-VI: Skilled, non-manual or manual, partly-skilled, unskilled occupations; [3] Missing information for 3 participants; [4] Missing information for 16 participants; [5] Missing information for 14 participants; [6] Missing information for 4 participants; [7] Missing information for 1 participant; [8] Missing information for 7 participants; [9] Excluding participants with COVID-19 symptoms before 6 March or after 31 March 2020; [10] No calculable only positive PCR cases recorded.

A non-adjusted comparison of subjects with a negative and a positive ECLIA is presented in Table 2. Negative ECLIA subjects were younger with a lower BMI and reported lower frequency of probably contacts with COVID-19 cases and MGEs attendance; shorter duration, no medical consultations, and full recovery COVID-19 disease. In contrast, one-third of the ECLIA positive subjects suffered some sequelae, not recovered completely and had a worse health level than before the COVID-19 disease. The ECLIA results did not differ by the O positive, ABO blood group, or vitamin D mean serum concentration.

We show in Table 3 the multivariate Poisson regression adjusted relative risk (aRR) of the association between the measured factors and an ECLIA negative result. The negative anti-SARS-COV-2 antibody group had a lower BMI (aRR = 0.83 95% CI 0.72–0.99), less probable contact with COVID-19 cases (aRR = 0.13 95% CI 0.03–0.51) and was younger (aRR = 0.96 95% CI 0.91–1.00).

Among the 484 participants, 469 (96.7%) had two anti-SARS-CoV-2 antibody determinations by ECLIA, one in the first study (June 2020) and the second in the current survey. We observed an overall decline in the values of the arbitrary units of the ECLIA test from June 2020, geometric mean 46.41 (95% CI 41.87–51.44) to October 2020, geometric mean 26.55 (95% CI 23.30–30.25) (*p* < 0.001). However, ninety-six participants (20.5%) had an increase of antibodies, whereas 373 participants (79.5%) had a decline. The comparison of the two groups is in Table 4. Participants with increased antibodies were older, not current smokers, had a higher BMI, a higher frequency of probable contact with COVID-19 cases, more medical consultation, and a longer duration of the illness. Vitamin D status and O ABO blood levels were similar in both groups.

After an adjusted Poisson regression analysis (Table 5), high BMI, older age, a medical consultation and more contact with COVID-19 cases were associated with an increase of anti-SARS-CoV-2 antibodies, being a current smoker was associated with a decline in anti-SARS-CoV-2 antibodies (Figure 2).

**Table 2.** Subjects' characteristic distribution by SARS-CoV-2 ECLIA result, and unadjusted relative risk of comparing negative versus persistent antibody (positive ECLIA) results. COVID-19 Borriana cohort, second seroepidemiological survey, October 2020.

| Variables | Negative Antibodies N = 5 (%) | Persistent Antibodies N = 479 (%) | RR [1] | 95% CI [2] | *p*-Value |
|---|---|---|---|---|---|
| Female | 4 (80.0) | 297 (62.0) | 0.41 | 0.05–3.66 | 0.426 |
| Age mean ± standard deviation | 24.0 ± 16.7 | 37.4 ± 17.1 | 0.95 | 0.90–1.01 | 0.106 |
| 0–24 years | 4 (80.0) | 139 (29.0) | | | |
| 25–44 | 0 | 157 (32.8) | | | |
| 45–64 | 1 (20.0) | 165 (34.4) | | | 0.123 |
| 65 and over | 0 | 18 (3.8) | | | |
| O ABO [3] | 4 (80) | 195 (40.8) | 5.71 | 0.64–50.8 | 0.116 |
| Vitamin D ng/mL | 28.2 ± 8.9 | 29.8 ± 9.2 | 0.98 | 0.87–1.09 | 0.687 |
| Body mass index [4] $Kg/m^2$ | 21.3 ± 2.7 | 25.0 ± 5.0 | 0.84 | 0.76–0.94 | 0.002 |
| <18.0 | 1 (20.0) | 40 (8.4) | | | |
| 18.0–24.9 | 4 (80.0) | 206 (43.3) | | | |
| 25.0–29.9 | 0 | 148 (31.1) | | | 0.038 |
| ≥30.0 | 0 | 85 (17.9) | | | |
| Occupation I-II [5] | 1 (20.0) | 144 (30.3) | 0.57 | 0.07–5.15 | 0.624 |
| Current smoker [6] | 1 (20.0) | 64 (13.8) | 1.19 | 0.37–3.82 | 0.776 |
| Physical exercise | 4 (80.0) | 285 (59.5) | 2.70 | 0.30–24.08 | 0.653 |
| Alcohol consumption [7] | 1 (20.0) | 107 (23.0) | 0.83 | 0.29–7.44 | 0.874 |
| Chronic illness [8] | 0 | 166 (34.9) | 0.28 | 0.00–2.06 | 0.240 |
| COVID-19 disease | | | | | |
| Family with COVD-1 9 case [9] | 2 (40.0) | 301 (63.0) | 0.39 | 0.07–2.35 | 0.308 |
| Probable contact COVID-19 case [10] | 2 (40.0) | 388 (82.2) | 0.15 | 0.03–0.88 | 0.044 |
| Assistance Mass Gathering Events ≥ 2 and over [11] | 1 (25%) | 255 (60.6) | 0.22 | 0.02–2.10 | 0.189 |
| Hospitalisations | 0 | 9 (1.9) | 7.85 | 0.00–19.23 | 1.000 |
| PCR positive | 0 | 26 (5.4) | 2.61 | 0.00–57.60 | 1.000 |
| Asymptomatic | 1 (20.0) | 53 (11.1) | 1.99 | 0.23–17.62 | 0.448 |
| Medical consultation | 0 | 208 (43.4) | 0.19 | 0.00–1.45 | 0.121 |
| Illness duration | 4.3 ± 6.7 | 10.4 ± 17.5 | 0.93 | 0.76–1.13 | 0.457 |
| Post-COVID-19 | | | | | |
| Sequelae | 0 | 159 (33.2) | 0.30 | 0.00–2.23 | 0.273 |
| Health as before the disease [12] | 5 (100.0) | 397 (83.1) | 1.36 | 0.18-∞ | 0.732 |
| Recover health [13] | 5 (100.0) | 390 (81.6) | 1.50 | 0.20-∞ | 0.800 |
| Exposure post-COVID-19 | | | | | |
| Social contact | 4 (80.0) | 103 (78.5) | 1.09 | 0.12–9.79 | 0.935 |
| Gathering people | 0 | 25 (5.2) | 2.73 | 0.00–20.04 | 1.000 |
| Trip out of Borriana [14] | 1(20.0) | 109 (22.9) | 0.85 | 0.10–7.50 | 0.880 |
| Restaurant assistance | 0 | 142 (29.6) | 0.36 | 0.00–2.63 | 0.352 |
| Terrace assistance [15] | 3 (60.0) | 286 (59.8) | 2.19 | 0.37–13.04 | 0.400 |

[1] RR = Relative risk; [2] CI = Confidence interval; [3] Missing information for 1 participant; [4] Missing information for 3 participants; [5] Missing information for 3 participants; [6] Missing information for 16 participants; [7] Missing information for 14 participants; [8] Missing information for 4 participants; [9] Missing information for 1 participant; [10] Missing information for 7 participants; [11] Excluding participants with COVID-19 symptoms before 6 March or after 31 March 2020; [12] Missing information for 1 participant; [13] Missing information for 1 participant; [14] Missing information for 2 participants; [15] Missing information for 1 participant.

**Table 3.** Adjusted relative risks (aRR) of factors associated with negative anti-SARS-CoV-2 antibody group versus persistent (positive ECLIA result), by Poisson regression. COVID-19 Borriana cohort 2020.

| Factors | aRR | 95% CI [1] | *p*-Value | Pearson Goodness of Fit |
|---|---|---|---|---|
| Body mass index (Kg/m$^2$) [2] | 0.87 | 0.77–0.99 | 0.037 | 0.817 |
| Age (years) [3] | 0.96 | 0.91–1.00 | 0.076 | 0.971 |
| Sex: Female [4] | 0.59 | 0.09–3.96 | 0.590 | 0.971 |
| O ABO blood group [5] | 5.52 | 0.61–49.57 | 0.127 | 0.971 |
| Occupation I-II [6] | 0.66 | 0.08–5.45 | 0.695 | 0.587 |
| Current smoker [7] | 2.43 | 0.14–41.87 | 0.541 | 0.072 |
| Physical exercise [7] | 3.05 | 0.36–25.71 | 0.305 | 0.999 |
| Alcohol consumption [7] | 1.07 | 0.10–11.77 | 0.954 | 0.398 |
| Chronic illness [8] | 0.42 | 0.00–3.35 | 0.456 | NC [9] |
| COVID-19 disease | | | | |
| Family with COVD-19 case [10] | 0.49 | 0.08–2.85 | 0.428 | 0.540 |
| Probable contact COVID-19 case [11] | 0.18 | 0.04–0.82 | 0.027 | 1.000 |
| Assistance Mass Gathering Events ≥ 2 and over [12,13] | 0.20 | 0.02–1.78 | 0.150 | 0.458 |
| Medical consultation [14] | 0.26 | 0.00–2.05 | 0.227 | NC [9] |
| Illness duration [15] | 0.94 | 0.77–1.15 | 0.570 | 1.000 |

[1] CI = Confidence interval; [2] Adjusted for age sex ABO, occupation current smoker physical exercise alcohol consumption; [3] Adjusted for sex ABO; [4] Adjusted for ABO age; [5] Adjusted for sex age. [6] Adjusted for age ABO sex; [7] Adjusted for age sex ABO occupation; [8] Adjusted for age sex ABO current smoker physical exercise alcohol beverage; [9] NC = No computable; [10] Adjusted for age sex ABO chronic illness current smoker physical exercise alcohol consumption assistance mass gathering events ≥ 2 and over; [11] Adjusted for age, sex ABO occupation current smoker physical exercise alcohol consumption assistance mass gathering events ≥ 2 and over; [12] Excluding participants with symptoms before 6 March or after 31 March 2020; [13] Adjusted for age, sex, ABO, occupation, current smoker physical exercise alcohol consumption assistance mass gathering events ≥ 2 and over; [14] Adjusted for age sex, ABO, chronic illness occupation. [15] Adjusted by age, sex, body mass index, ABO, chronic illness.

**Table 4.** Comparison between participants with increase and decline anti-SARS-CoV-2 antibodies, considering the first and second SARS-CoV-2 serologic surveys. COVID-19 Borriana cohort 2020.

| Variables | Increase SARS-CoV-2 N = 96 (%) | Decline SARS-CoV-2 N = 373 (%) | *p*-Value |
|---|---|---|---|
| Female | 60 (62.5) | 236 (63.5) | 0.906 |
| Age mean ± standard deviation | 45.3 ± 18.0 | 34.9 ± 16.3 | 0.001 |
| 0–24 years | 15 (15.6) | 126 (33.8) | |
| 25–44 | 24 (25.0) | 131 (35.1) | 0.000 |
| 45–64 | 47 (49.0) | 108 (29.0) | |
| 65 and over | 10 (10.4) | 8 (2.1) | |
| O ABO [1] | 45 (46.9) | 149 (40.1) | 0.246 |
| Vitamin D ng/mL | 29.4 ± 9.0 | 30.0 ± 9.3 | 0.621 |
| Body mass index (Kg/m$^2$) | 26.9 ± 5.0 | 24.5 ± 4.9 | 0.001 |
| <18.0 | 6 (6.3) | 38 (10.2) | |
| 18.0–24.9 | 27 (28.1) | 176 (47.2) | 0.001 |
| 25.0–29.9 | 35 (36.5) | 107 (28.7) | |
| ≥30.0 | 38 (40.0) | 55 (14.7) | |
| Occupation I-II [2] | 29 (30.2) | 107 (28.9) | 0.802 |
| Current smoker [3] | 7 (7.5) | 56 (15.6) | 0.003 |
| Physical exercise | 57 (59.4) | 222 (59.5) | 1.000 |
| Alcohol beverages [4] | 24 (25.8) | 82 (22.7) | 0.583 |
| Chronic illness [5] | 41 (43.2) | 123 (45.6) | 0.092 |
| COVID-19 | | | |
| Family with COVD-19 case [6] | 63 (65.6) | 230 (61.8) | 0.555 |
| Probable contact COVID-19 case [7] | 83 (89.2) | 294 (79.7) | 0.036 |
| Assistance Mass Gathering Events ≥2 and over [8] | 61 (63.5) | 227 (60.9) | 0.724 |
| Hospitalisations | 3 (3.1) | 6 (1.6) | 0.398 |
| PCR positive | 8 (8.3) | 17 (4.6) | 0.198 |
| Asymptomatic | 7 (7.3) | 46 (12.3) | 0.206 |
| Medical consultation | 56 (58.3) | 144 (38.6) | 0.001 |
| Illness duration | 12.0 ± 15.0 | 10 ± 18.2 | 0.034 |

[1] Missing information for 1 participant; [2] Missing information for 3 participants; [3] Missing information 16 participants; [4] Missing information for 11 participants; [5] Missing information for 3 participants; [6] Missing information for 1 participant; [7] Missing information for 7 participants; [8] Excluding participants with COVID-19 symptoms before 6 March or after 31 March 2020.

**Table 5.** Adjusted relative risks (aRR) of factors associated with an increase of anti-SARS-CoV-2 antibodies, considering the first and second SARS-CoV-2 serologic surveys, by Poisson regression. COVID-19 Borriana cohort 2020.

| Variables | aRR | 95% CI [1] | *p*-Value | Pearson Goodness of Fit |
|---|---|---|---|---|
| Body mass index (Kg/m$^2$) [2] | 1.06 | 1.02–1.10 | 0.001 | 0.996 |
| Age (years) [3] | 1.03 | 1.02–1.04 | 0.000 | 0.991 |
| Current smoker [4] | 0.48 | 0.24–0.96 | 0.037 | 0.997 |
| Probable contact COVID-19 case [5] | 2.04 | 1.12–3.74 | 0.022 | 0.994 |
| Medical consultation [6] | 1.62 | 1.12–2.34 | 0.010 | 0.991 |
| Illness duration [7] | 0.99 | 0.99–1.01 | 0.543 | 0.966 |

[1] CI = Confidence interval; [2] Adjusted for age sex ABO, occupation current smoker physical exercise alcohol consumption; [3] Adjusted for sex ABO; [4] Adjusted for age sex ABO occupation; [5] Adjusted for age, sex ABO occupation current smoker physical exercise alcohol consumption assistance mass gathering events ≥ 2 and over; [6] Adjusted for age sex, ABO, chronic illness occupation; [7] Adjusted by age sex, body mass index, ABO, chronic illness.

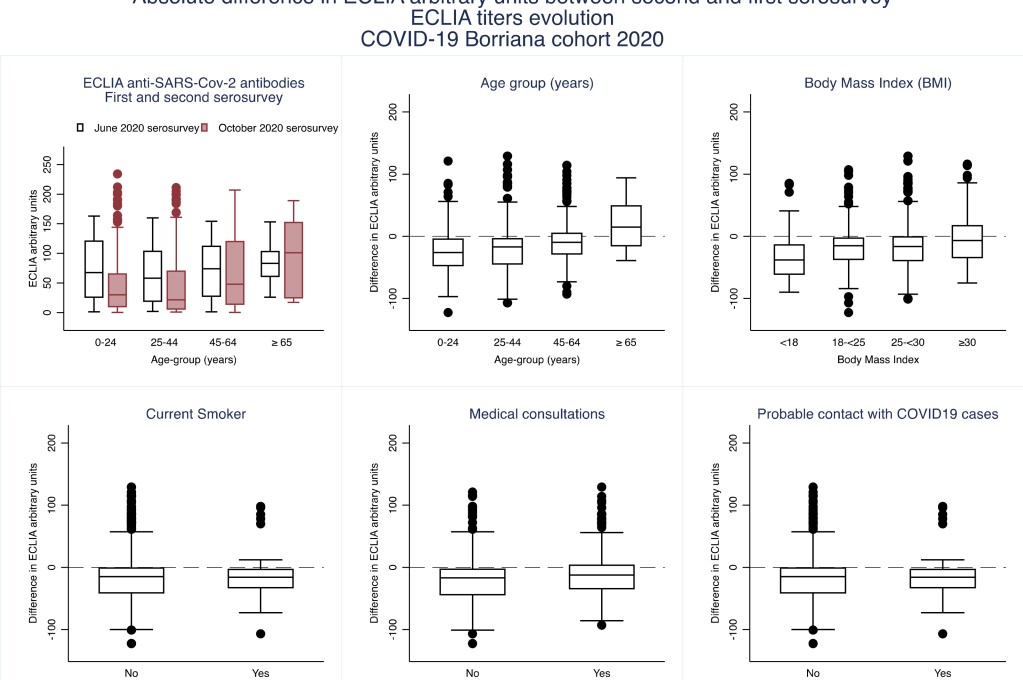

**Figure 2.** The absolute difference in ECLIA arbitrary units between second and first serosuvey. ECLIA titers evolution COVID-19 Borriana cohort 2020.

## 4. Discussion

Our results show that 99% (479/484) of COVID-19 patients in the cohort maintained anti-SARS-CoV-2 antibodies in the first six months after the COVID-19 episode. We did not find evidence of any SARS-CoV-2 reinfection episode in this group in the period between the first and second serosurvey.

The factors associated with a negative result in the second serosurvey were a lower BMI, lower frequency of probable contact with COVID-19 cases, younger age and a milder COVID-19 episode when compared to the group with antibody persistence. Besides, we observed a significant overall decline in anti-SARS-CoV-2 antibodies. However, a fifth of participants showed an increase in anti-SARS-CoV-2 antibodies, which was probably related to the infection course or the frequency of contacts with other cases.

The persistence of anti-SARS-CoV-2 antibodies after the infection, measured by different serologic techniques, including ECLIA [17,18] or other methods [19–22], has been reported to last between 4 and 8 months. However, significant differences in antibodies persistence associated with the detection techniques have been found, and total antibodies against S1 and the S1 receptor-binding domain (RBD) were detected for longer [23].

Several studies have described a similar decline of neutralising SARS-CoV-2 antibodies. This finding supports the need for follow-up studies to gather evidence on the duration of the antibody response and the protection against reinfections [24,25]. Although we did not observe reinfection episodes in our cohort, others have reported reinfection episodes. A 96% protective effectiveness in prior infected patients was estimated in care homes outbreaks [26,27].

In the United States, and from data generated from commercial laboratory analysis of SARS-COV-2 antibody tests of 3.2 million patients, Harvey et al. found 0.30% reinfections [28]. In the United Kingdom, two cohort studies of health care workers comparing positive anti-SARS-COV-2 antibody group versus negative group during six months of follow-up [29], have found reinfection rate ratios of 0.11 and 0.16, respectively [30]. In a vessel outbreak, three crew members with previous positive anti-SARS-COV-2 antibodies had not suffered infection versus 104 of 122 (85.2%) crew members with negative antibodies, who suffered a SARS-CoV-2 infection ($p = 0.002$) [31].

A more severe COVID-19 disease is usually associated with higher anti-SARS-CoV-19 antibodies [32–34], and antibodies correlate with the duration of the infection, older age and hospitalisation [35,36]. Still, in other studies, age and symptoms were not associated with antibody levels [37], suggesting high variability. Also, high levels of anti-SARS-CoV IgG and neutralising antibodies were observed in COVID-19 patients with high BMI and patients with metabolic syndrome [33,38–40]. These results are in line with our finding, where low BMI was associated with negative antibodies. However, in obese COVID-19 patients, a reverse association has been observed between high BMI and lower SARS-CoV-2 antibodies [41]. In addition, the increase of SARS-CoV-2 antibodies in the second survey was associated with age older, higher BMI, more severe disease [42] and low current smoke. A hypothesis was suggested that a tobacco mosaic RNA virus, presented in the respiratory tract of smokers, could have some protection against the SARS-COV-2 virus [43]. However, the harmfulness of tobacco smoking in COVID-19 patients must be stressed in conjunction with continuing the research.

Factors indicating less exposure to COVID-19 cases in the group with negative antibodies suggest that the intensity, type and duration of exposure could play a role in developing the disease and its course [37,44,45]. Then, high viral load exposure with high duration may be decisive for the development and course of COVID-19 infection [46]. These results emphasised the crucial preventive measures to stop COVID-19 transmission: Keep distance, use masks, hand-washing and reduce time and contact of potential exposures.

The O ABO blood group was not associated with negative antibody groups, in line with the finding of Wendel et al. [38]. Still, the small sample of the negative antibody group prevents a definitive conclusion. Moreover, Vitamin D status was not related to the negative antibody group, but the measure as prevalence excludes a potential role [47]. On the other hand, negative antibodies were not associated with sex in contrast with Markmann et al.'s work [33], where men had more neutralising antibody levels than female.

Serologic surveys of SARS-CoV-2 infection are a useful approach in the control and prevention of COVID-19 disease, including the surveillance of the disease, to characterise the effectiveness of vaccinations against the disease, the antibody response relation to disease course, and the factors associated with the duration of the immune response [3].

Many serologic tests of SARS-CoV-2 measure IgA, IgG and IgM, with high heterogeneity in sensitivity and specificity [48]. The ECLIA technique has a good accuracy [3,49,50] and has been recommended for population screening [51]. Anti-SARS-CoV-2 antibodies obtained by ECLIA positively correlate with neutralising antibodies, but their sensitivity should be improved [52,53].

Our study has some strengths; first, a prospective cohort design; second, the elevated participation rate; third, the use of a serologic test with reasonable sensitivity and specificity; four, the use of multivariate analysis to control potential confounding; fifth, a follow-up of up to 6 months; and finally, a population-based approach.

As limitations, the quantitative ECLIA anti-SARS-CoV-2 antibodies test that we used has a significant, but feeble correlation (Pearson's correlation coefficient of 0.37 $p < 0.001$ and poor linear relationship with the enzyme-linked immunosorbent assay (ELISA AESKULISA [R]) [54]. The ECLIA test does not measure the response to the S antigen of SARS-CoV-2, but detects antibodies against the nucleocapsid N. The non-simultaneous analysis of the blood samples from the two studies delimits the results, but only a qualitative approach, increase versus decline, was used.

Finally, the small number of participants with negative anti-SARS-CoV-2 antibodies decreases the power to detect associated factors; the younger age, and in general, milder COVID-19 disease of the cohort does not permit a generalisation of results, and we cannot discard the existence of unknown factors that we did not consider in this new disease.

In subsequent studies, a quantitative determination of neutralising antibodies against RDB antigen [55,56] could be helpful to study the immunity against the SARS-CoV-2 virus considering the importance to define correlate levels of protective immunity [57,58] and more precise estimation of potential factors associated with the clinical course [59,60] As we can assume the future existence of vaccinated participants against SARS-CoV-2 in the cohort, the follow-up of the cohort would provide information on the response after vaccination against SARS-CoV-2 in patients who have suffered the infection [61] and inform the need of additional vaccine doses.

## 5. Conclusions

In the first six months after a COVID-19 infection, a high proportion of participants maintained detectable anti-SARS-CoV-2 antibodies, and we did not observe new COVID-19 episodes in the follow-up period.

**Author Contributions:** Conceptualization, S.D.-M., A.A.-P., M.R.P.-S., L.G.-L., D.J.-S., L.A.-E., U.C.-A., S.F.-R., M.S.-U.; methodology, A.A.-P.,S.D.-M., J.P.-B., P.V.-U., M.L.-P., A.D.R.-G., S.F.-R., M.S.-U., G.F.-A., L.A.-E., G.B.-M., U.C.-A., C.D.-B., M.M.-B.; software A.A.-P., M.R.P.-S., J.P.-B.,M.C.L.-D.; validation, J.P.-B., C.N.-R. M.R.P.-S.; formal analysis, M.R.P.-S., A.A.-P., J.P.-B., S.D.-M.; investigation, S.D.-M., M.R.P.-S., P.V.-U., M.L.-P., A.D.R.-G., S.F.-R., G.F.-A., M.S.-U., L.A.-E., G.B.-M., B.C.-F., U.C.-A., C.D.-B., M.F.-C., L.G.-L., D.J.-S., M.C.L.-D., M.D.L.-V., M.M.-B., C.N.-R., R.R.-P., S.V.-L.; resources, S.D.-M., G.F.-A., G.B.-M., B.C.-F., M.F.-C., L.G.-L., M.C.L.-D., M.D.L.-V., M.L.-P., C.N.-R., R.R.-P., S.V.-L.; data curation, A.A.-P., J.P.-B., P.V.-U., M.L.-P., A.D.R.-G., M.R.P.-S.; writing original draft preparation, A.A.-P., J.P.-B., M.R.P.-S., M.C.L.-D., S.D.-M., S.F.-R.; writing-review and editing, J.P.-B., A.A.-P., S.D.-M., U.C.-A., M.F.-C., M.C.L.-D.; visualization, J.P.-B., D.J.-S., L.A.-E., L.G.-L.; supervisión M.R.P.-S., M.C.L.-D.; project admistration, S.D.-M., A.A.-P., M.C.L.-D., G.F.-A.; funding acquision, S.D.-M., M.C.L.-D., A.A.-P. All authors have read and agreed to the published version of the manuscript.

**Funding:** This research received no external funding.

**Institutional Review Board Statement:** The study was conducted according to the guidelines of the Declaration of Helsinki, and the study of this cohort was a part of the public health surveillance as a prolongation of the COVID-19 outbreak in the Falles festival of Borriana control measures [62], and was exempt from the Ethics Review Board approval's protocol according to the Spanish legislation, including the General Low including the General Law of Health [63], the Law of Cohesion and Quality of the National System of Health [64], and the Law General of Public Health [65]. In addition, the cohort was followed to respond to a new disease, the COVID-19 pandemic [66].

**Informed Consent Statement:** Informated consent was obtained from all subject involved in the study.

**Data Availability Statement:** Data of the this study can be consulted if the authors are requested. Dataset: borrianacohort.dta.

**Acknowledgments:** We thank the participants of the cohort and the Borriana's *Falles* organisation for the support that made it possible to perform this study. In addition, we appreciate the assistance and support of Roser Blasco-Gari, Helena Buj-Jorda, Israel Borras-Acosta, Lucia Castell-Agusti, Mercedes De Francia-Valero, Maria Domènech-Molinos, Marc Garcia, Maria Gil-Fortuño, Elena Grañana-Toran, Noelia Hernández-Perez, Laura Lopez-Diago, Salvador Martinez-Parra, Sara Moner-Marin, Silvia

Pesudo-Calatayud, Lara Sabater-Hernández, Maria Luisa Salve-Martinez, Irene Suarez-Linares, Juan José Ventura-Buchardo, and Alberto Yagüe-Muñoz to carry out the study.

**Conflicts of Interest:** The authors declare no conflict of interest.

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
