# Peer review of "Persistence of Anti-SARS-CoV-2 Antibodies Six Months after Infection in an Outbreak with Five Hundred COVID-19 Cases in Borriana (Spain): A Prospective Cohort Study"

_covid, doi:10.3390/covid1010006_

Round 1
Reviewer 1 Report
INTRODUCTION: a prominence should be given to the fact that the antibodies level is not an exclusive item to be considered to know the duration of immunity after COVID-19, referring to the relevance of the cellular immunity (e.g. estimated by means of the numbers of virus-specific T-cells).
Excellent analysis of the possible impact of refusing an invitation to enter the study.
Excellent design and discussion, but a brighter evidence should be given (inside the conclusions or, if preferred, inside the discussion itself) to the practical spin-off of some definite results looking for appropriate public health strategies. The evidence for adverse effects of current smoking should become a fundamental component of the risk communication for public health services and contact tracers. An interpretation should be given regarding the determinants of the effect of the numbers and the characteristics of contacts with COVID-19 cases, reasonably accounting for different levels (duration, intensity) of the exposure.

Author Response
Castelló de la Plana, June 16, 2021.
Editors of COVID
Dear Ms. Biljana Savic and Ms. Wynne Wang:
We attach the revision of our manuscript” Manuscript ID: covid-1248809:
Title: Persistence of anti-SARS-CoV-2 antibodies six months after infection
in an outbreak with five hundred COVID-19 cases in Borriana (Spain): A
prospective cohort study. Authors: Salvador Domènech-Montoliu et al.
Thank you very much for the revision and comments of the two reviewers.
We appreciated your effort to achieve a very rapid revision.
We have followed the indications of the reviewers.
Response to the first reviewer:
Thank you very much for your revision and for your comments.
- INTRODUCTION: a prominence should be given to the fact that the antibodies level is not an exclusive item to be considered to know the duration of immunity after COVID-19, referring to the relevance of the cellular immunity (e.g. estimated by means of the numbers of virus-specific T-cells).
We add an indication of the cellular immunity on the COVID-19, and two references have been included.
- Excellent analysis of the possible impact of refusing an invitation to enter the study.
Thank you for this remark.
- Excellent design and discussion, but a brighter evidence should be given (inside the conclusions or, if preferred, inside the discussion itself) to the practical spin-off of some definite results looking for appropriate public health strategies.
We appreciate your comments and following your indications, the two points are addressed.
3.1. The evidence for adverse effects of current smoking should become a fundamental component of the risk communication for public health services and contact tracers.
An indication of the harmfulness of tobacco smoking is added and the opportunity for tobacco research is contemplated.
3.2 An interpretation should be given regarding the determinants of the effect of the numbers and the characteristics of contacts with COVID-19 cases, reasonably accounting for different levels (duration, intensity) of the exposure.
The importance of preventive measures like distance, hand-washing, and mask protection is indicated.
Response to the second reviewer:
Thank you very much for your revision and for your comments.
We address the comments of the reviewer.
- I think that the comparison before-after for the 469 subjects who gave two blood samples should have been made examining both samples simultaneously (which means re-examining the first samples together with the second ones), but I do not know whether the first samples were still available).
The first samples were not available to compare with the second samples in the second study. We had indicated this situation in the methods and in the discussion as a limitation of the study.
- I am surprised that the subjects who showed persistence in antibodies had a higher BMI, while I am not surprised that they had higher frequency of probable contact with COVID-19 cases and had a longer duration of their illness
At this moment there is a controversy around BMI and humoral immunity in COVID-19 patients. In general, the levels of anti-SARS-CoV-2 antibodies are high when body mass index is high but the research continues. We included studies with opposing results (1-4). On the other hand, an increase of exposure corresponds to high anti-SARS-CoV-2 antibody levels.
1.Wendel S, et al. A longitudinal study of convalescent plasma (CCP) donors and correlation of ABO group, initial neutralizing antibodies (nAb), and body mass index (BMI) with nAb and anti-nucleocapsid (NP) SARS-CoV-2 antibody kinetics: Proposals for better quality of CCP collections. Transfusion. 2021;61:1447-1460.
- Racine-Brzostek SE, et al. Postconvalescent SARS-CoV-2 IgG and neutralizing antibodies are elevated in individuals with poor metabolic health. J Clin Endocrinol Metab. 2021;106:e2025-e2034.
3.Gerhards C, et al. Longitudinal assessment of anti-SARS-CoV-2 antibody dynamics and clinical features following convalescent from COVID-19 infection. Int J Infect Dis. 2021:S1201-9712(21)00392-1.
- Frasca D, et al. Influence of obesity on serum levels of SARS-CoV-2-specific antibodies in COVID-19 patients. PLoS One. 2021;16:e0245424.
Best wishes from Castelló de la Plana
Alberto Arnedo-Pena, on behalf of the authors.

Reviewer 2 Report
I think that the comparison before-after for the 469 subjects who gave two blood samples should have been made examining both samples simultaneously (which means re-examining the first samples together with the second ones), but I do not know whether the first samples were still available).
I am surprised that the subjects who showed persistence in antibodies had a higher BMI, while I am not surprised that they had higher frequency of probable contact with COVID-19 cases and had a longer duration of their illness
Author Response
see text
